# Necrotizing Laryngitis in Patients with Hematologic Disease: The First Case-Report Due to PDR *Acinetobacter baumannii* and Literature Review

**DOI:** 10.3390/microorganisms12071382

**Published:** 2024-07-08

**Authors:** Ioanna Tatouli, Nikolaos Dedes, Andreas Bozikas, Stamatoula Melliou, Maria-Markella Pavlou, Sofoklis Kontogiannis, Efthymios Kyrodimos, Eftychia Kanioura, Ioannis Ntanasis-Stathopoulos, Meletios-Athanasios Dimopoulos, George Dimopoulos, Efstathios Kastritis, Maria Gavriatopoulou

**Affiliations:** 1Department of Clinical Therapeutics, School of Medicine, National and Kapodistrian University of Athens, 11528 Athens, Greece; ioannatatouli@gmail.com (I.T.); 90andreas@windowslive.com (A.B.); matoula92@gmail.com (S.M.); maria-markella_93@live.com (M.-M.P.); agupiela@gmail.com (S.K.); mdimop@med.uoa.gr (M.-A.D.); gdimop@med.uoa.gr (G.D.); ekastritis@med.uoa.gr (E.K.); 2First Department of Otolaryngology, Hippocration General Hospital, School of Medicine, National and Kapodistrian University of Athens, 11528 Athens, Greece; ekirodim@med.uoa.gr (E.K.);

**Keywords:** necrotizing laryngitis, *Acinetobacter baumannii*, multiple myeloma, hematologic disease, hemophagocytic lymphohistiocytosis

## Abstract

Immunocompromised patients with hematologic diseases may experience life-threatening infections with rather uncommon manifestations. Laryngitis has been described as a potential infection in such vulnerable patients and may result in major complications, ranging from impending airway obstruction to total laryngeal necrosis. Immediate laryngoscopy is of paramount importance, as it provides quantification of laryngeal edema and evidence of necrosis. Documentation of the causative pathogen is usually feasible through tissue culture. In the literature, 14 cases of necrotizing laryngitis have already been published. Here, we present the case of a 38-year-old male with a recent diagnosis of multiple myeloma, who received the first cycle of therapy a few days before admission. The patient presented with neutropenic fever, diarrhea, and multiple organ dysfunction. His course was complicated with hemophagocytic lymphohistiocytosis and stridor. A diagnosis of necrotizing laryngitis attributed to *Acinetobacter baumannii* invasion of the larynx was established. This manuscript highlights that the management of patients with hematologic disease and necrotizing laryngitis should be coordinated in highly specialized centers and clinicians should have a high level of clinical suspicion and act promptly.

## 1. Introduction

Infections represent a major cause of severe morbidity and mortality in immunocompromised patients, particularly those who suffer from hematologic diseases. A wide diversity of microorganisms, including common community-acquired bacteria and viruses and uncommon opportunistic pathogens, have been incriminated in such serious and life-threatening infections that may be localized in quite unusual sites such as the larynx [1]. Laryngitis in the immunocompromised host may rapidly lead to extreme laryngeal edema and/or necrosis, and for patients with impending complete airway obstruction, securing the airway should be the first priority [2].

Among hematologic malignancies, multiple myeloma primarily affects patients in the seventh decade of life [3]. Infections usually complicate such patients because of the immunosuppressive status of disease and immunomodulatory treatments. During the first months after diagnosis, the incidence of serious bacterial (predominantly *Streptococcus pneumoniae*, *Staphylococcus aureus*, *Enterobacteriaceae*) and viral infections (mainly *Varicella-Zoster Virus*—particularly in the context of proteasome inhibitor administration) is increased 10-fold in comparison with healthy controls [4,5]. Although most infections are due to common community-acquired pathogens, the increased survival and prolonged hospitalizations of patients with multiple myeloma have led to an increased frequency of nosocomial causative agents. In this manuscript, we present the case of a 38-year old male, who experienced a series of rare and life-threatening complications of multiple myeloma presenting initially with hemophagocytic lymphohistiocytosis (HLH), whose course was complicated by necrotizing laryngitis due to pan-drug-resistant (PDR) *Acinetobacter baumannii*.

## 2. Case Description

The patient was a 38-year-old man with an unremarkable prior medical history. Due to progressively worsening fatigue, he sought medical evaluation that revealed normocytic, normochromic anemia, with elevated erythrocyte sedimentation rate and increased levels of serum total protein. Serum protein electrophoresis recorded the presence of monoclonal IgGκ, and subsequently a bone marrow biopsy noted 95% infiltration by monoclonal plasma cells. Treatment was initiated immediately with bortezomib, lenalidomide, dexamethasone, and daratumumab. A few days thereafter, erythroderma appeared and the possibly culpable agent lenalidomide was discontinued. However, the patient soon developed fever up to 40 °C without rigors, dyspnea, abdominal pain, and dysuria, and thus was admitted to our hospital. Laboratory and imaging studies were consistent with multiple organ dysfunction syndrome. Anemia, thrombocytopenia, markedly increased serum ferritin and inflammation markers, hepatocellular necrosis, acute kidney injury with decreased urine production and metabolic acidosis, hepatosplenomegaly, and reduced left ventricular ejection fraction were noticed. Clinically, the patient was in an acute confusional state but was hemodynamically stable, with no need for oxygen supply. Subsequently, admission to the high dependency unit (HDU) was decided and broad-spectrum antibiotics (adjusted dose of meropenem and vancomycin) were administered, the patient was also put on continuous venovenous hemodiafiltration, and extensive diagnostic workup was undertaken (laboratory values upon admission are shown in Table 1). Nonetheless, the patient’s laboratory profile deteriorated rapidly and therefore HLH was also included in the differential diagnosis of the underlying cause. The HScore (a scoring system predictive of the possibility of HLH) suggested with 99% certainty the presence of HLH, and chemotherapy consisting of etoposide and dexamethasone was started on the second day of his admission to the HDU. The treatment regimen appeared to be effective, as the patient displayed clinical and laboratory signs of improvement and was eventually weaned off dialysis on the 5th day of admission to the HDU. Neutrophil count fell below 1000/μL the following day but was otherwise asymptomatic, and antibiotics were thus discontinued on day 10.

The patient’s course remained stable until day 15, when he developed fever with rigors. At this timepoint, he was on the 9th day of neutropenia, and therefore an antibiotic scheme consisting of meropenem, amikacin, colistin, and daptomycin was administered, while a relative investigation was undertaken with the intent to identify the causative microorganism. The patient had severe neutropenia and thrombocytopenia (Neutrophils: 0.1 × 10^3^/μL, Platelet count: 17 × 10^3^/μL). A few hours thereafter, he presented with progressively worsening hoarseness. He had the characteristic “hot potato voice” and lung auscultation was notable for stridor and wheezing. Chest X-ray was normal. However, impending upper airway obstruction was suspected and laryngeal edema was subsequently confirmed by cervical computed tomography. Dyspnea worsened and arterial blood gas parameters indicated respiratory acidosis. In this context, it was decided to intubate the patient, but endotracheal tube placement was impossible due to the presence of a mass protruding towards the airway lumen from the left hemilarynx. Consequently, emergency percutaneous tracheostomy was successfully performed.

Shortly after the procedure was concluded, a series of blood cultures obtained at the time of the recent fever spike resulted positive for PDR *Acinetobacter baumannii*, and the antibiotic treatment was altered to ampicillin/sulbactam, meropenem, and colistin, as in vitro tests suggested synergistic antimicrobial effectiveness. Sputum culture that was obtained following tracheostomy did not yield growth of any pathogen (the patient was on broad spectrum antibiotics), yet molecular testing was positive for *Acinetobacter baumannii*. The patient was soon liberated from the ventilator and was referred for excision of the abnormal laryngeal tissue via panendoscopy (Figure 1a–c). Histopathological examination (Figure 1d) of the obtained specimen was consistent with necrotizing laryngitis, while PDR *Acinetobacter baumannii*—the same as in the blood cultures—was isolated from tissue cultures. In this context, the patient received the aforementioned antibiotic regimen for two months. During this time period, he also resumed anti-myeloma treatment (bortezomib, cyclophosphamide, dexamethasone, daratumumab). The patient was dismissed after 77 days of hospitalization.

Subsequently, the patient had a relatively uneventful course. Tracheostomy tube was displaced a few weeks after discharge, and following administration of six cycles of anti-myeloma treatment, the patient successfully received mobilization with cyclophosphamide, conditioning with melphalan, and underwent autologous stem cell transplantation with a full hematologic recovery after 10 days. One year after the myeloma diagnosis, the disease remains in complete response and the patient has ECOG performance status 0 with close to baseline functionality.

## 3. Review of Laryngeal Infections in Patients with Hematologic Diseases

Publicly available databases were reviewed for case-series and case-reports of patients with laryngeal infection in the context of hematologic disease. The search yielded 52 cases, the majority of which (39/52) regarded patients ≥16 years of age, while the remainder were cases of infants and children. The median age of the patients was 41 years (ranging from 1.5 to 74 years) and 26 out of 52 were female. In order of decreasing frequency, the underlying conditions were acute myeloid leukemia (16 cases), lymphoma (12 cases), acute lymphocytic leukemia (9 cases), aplastic anemia (4 cases), chronic lymphocytic leukemia (3 cases), multiple myeloma (2 cases), hemophagocytic lymphohistiocytosis (2 cases), and miscellaneous causes (4 cases—e.g., autologous stem cell transplant). Overall, 14 cases (6–19) displayed tissue necrosis, either through laryngoscopy or histologic examination exhibiting characteristics of necrotizing laryngitis. Table 2 summarizes the important attributes of the relevant cases, such as demographic characteristics, the responsible pathogen, the need to secure the airway, and patient outcomes for both those with necrotizing and non-necrotizing laryngeal infection.

Laryngoscopic evaluation of the patients seemed to reveal some particular characteristics of the laryngeal infection regarding the extent, the exact localization, and the appearance of the lesions. Among the 38 cases without evidence of laryngeal necrosis, acute epiglottitis was noted in 21 of them. In such cases, common findings were local inflammation (reddish color and edema of epiglottis), which may be accompanied by other lesions like ulcers, white or grayish thrush, pseudomembranes, or a protruding mass of the epiglottis. In this group, the leading pathogens were *Candida* spp. (43%), followed by the usual upper-respiratory infectious agents (33%)—e.g., *Streptococcus pneumoniae*, *Moraxella catarrhalis*, etc. Furthermore, there were some cases in which other findings were visualized, such as vocal cord ulcers in the absence of edema. Laryngoscopic image consistent with necrotizing epiglottitis was mentioned in 5 out of the 14 cases of patients with necrotizing infection. In the majority of these cases, cultures revealed poly-microbial growth with at least two pathogens being isolated (e.g., *Aspergillus*, *E. coli*, etc.).

Review of the microbiology data led to some important ascertainments. In both groups, fungi were the most frequent associated pathogen. Although *Candida* spp. predominate in the non-necrotizing group (16 out of 22 reports of fungal infection in this group), *Aspergillus* spp. were usually associated with necrotizing laryngitis (8 out of 10 reports of fungal infection in this group). Furthermore, 26% of documented pathogens in patients with necrotizing infection were Gram-negative bacteria, some of which displayed intrinsic or acquired resistance to common antibiotics. Moreover, regardless of the presence of necrosis, laryngeal infection may be due to multiple causative agents and only rarely were cultures negative. Among patients with necrotizing laryngeal infection (Table 3), emergency airway management was required quite often (85% of cases). Five of these patients died (3 due to aspergillosis and 1 due to mucormycosis). The remaining nine patients had progressive improvement of their status and were eventually discharged from hospital.

## 4. Discussion

Patients with hematologic disease are prone to severe infections. In particular, patients with multiple myeloma are de facto immunosuppressed as the malignant B-cell clone displaces other B-cell populations, which leads to defective humoral immunity, despite the abundance of gamma globulins. Furthermore, lymphocytopenia and neutropenia due to bone marrow invasion additionally hamstring immunity, while cytokines excreted by malignant cells (e.g., IL-6, IL-10, TGF-b) impair T-cell response to infection. Moreover, renal failure attributable to the disease or its treatment and iron overload secondary to transfusion therapy carry a supplementary predisposition to infection. In conjunction with these fundamental characteristics of the disease, further immunosuppression is accumulated because of the multiple successive lines of treatment that these patients are administered. The advent of novel agents, as well as autologous hematopoietic-stem cell transplantation (HCT), has widened the infectious “repertoire” to include a multitude of viral and fungal pathogens in parallel with the traditional capsulated microorganisms [20]. The relatively rare possibility of coexistence with hemophagocytic lymphohistiocytosis aggravates the already existing immunosuppression of multiple myeloma [21].

Laryngeal infection and epiglottitis in particular have long been considered medical emergencies and the treatment strategy often requires immediate airway management due to impending blockage [22]. Laryngeal necrosis or necrosis of the components of the laryngeal apparatus (such as the epiglottis or the glottis) can lead to catastrophic outcomes. The first case report of necrotizing laryngitis was described in 1983. In this case, a 21-year old female, who was diagnosed with acute lymphocytic leukemia and was undergoing chemotherapy, developed hoarseness, a sore throat, and respiratory distress. Laryngoscopy revealed necrotizing epiglottitis, and *Aspergillus* spp. was isolated from pharyngeal swab cultures. Despite treatment with amphotericin B, she underwent tracheostomy. Follow-up endoscopy noted complete necrosis of the epiglottis, while tissue cultures yielded growth of *Aspergillus* and *Klebsiella pneumoniae*. Notwithstanding all measures, the patient developed septic shock and succumbed.

Necrotizing laryngeal infection has been described in immunocompetent hosts as well. Recent reports in the literature include the case of a patient with necrotizing laryngitis in the context of *Aspergillus fumigatus* infection, whose personal history was notable only for well-managed diabetes [23]. Moreover, there is another case of a 50-year-old woman with an unremarkable past medical history, who developed necrotizing epiglottitis. Cultures yielded *Beta hemolytic streptococcus*, *Alpha hemolytic streptococcus*, *Neisseria elongate*, and *Candida albicans* [24]. Furthermore, Anastasiadou et al. [25] described a patient with acute necrotizing and ulcerative epiglottitis, also known as “black larynx”, whose blood cultures were positive for *Acinetobacter* spp. This patient’s sole comorbidity was arterial hypertension. There are also older reports of immunocompetent hosts with necrotizing laryngeal infections in association with infectious mononucleosis [26], group A *Streptococcus* [27], *Serratia marcescens* [28], and most interestingly a case of acute epiglottitis, which resulted in a retropharyngeal abscess and descending necrotizing mediastinitis [29].

However, necrotizing laryngitis is far more common in immunocompromised patients, such as was the case of our patient with recently-diagnosed multiple myeloma and concurrent hemophagocytic lymphohistiocytosis. Aside from hematologic diseases, there are mentions of laryngeal necrosis in the wider context of immunosuppressive situations. Relative to this, patients with laryngeal cancer (particularly when irradiation therapy has been applied) may be subject to perilesional necrotizing infections [30,31,32]. Associated pathogens include *Actinomyces*, *Aspergillus* spp., and *herpes simplex virus*. Moreover, especially vulnerable to such infections are deeply immunosuppressed patients, such as transplant recipients and those afflicted from HIV. *Staphylococcus aureus*, *Actinomyces*, *Pseudomonas aeruginosa*, *CMV*, *Aspergillus* spp., and *Candida* spp. have been reported as causative agents in these patients [33,34].

Particularly in patients with hematologic disease, direct laryngoscopy is of paramount importance to making a correct diagnosis, while tissue cultures should not be omitted, as this may act as a guide for targeted antibiotic treatment. Regarding our patient, immediate laryngoscopy showed a yellowish area beneath the left true vocal cord that was assessed as a possible necrotic lesion. The observation of necrosis during laryngoscopy should always raise the suspicion of an opportunistic infection. At the same time, hospital-acquired microorganisms cannot be excluded. In this instance, broad spectrum antibiotics should be administered, while microbial carriage and the possibility of multi-drug resistant and opportunistic pathogens should always be taken into account. Of the latter, fungi (especially *Aspergillus* spp.) are common causative agents of necrotizing infections among hematologic patients, which are frequently associated with need for intubation and increased mortality.

The literature review revealed only two cases of laryngitis in patients with multiple myeloma [35,36]. In both of them, the culpable microorganism was *Streptococcus pneumoniae*, the infection was non-necrotizing, and the outcome was good. Moreover, a further two cases of patients with hemophagocytic lymphohystiocytosis [7,13] developed severe necrotizing laryngitis (one was due to non-fungal, poly-microbial infection and the other due to *Pseudomonas aeruginosa*), eventually requiring intubation or tracheostomy. To our knowledge, there has been no other report of a patient with necrotizing laryngitis with a background of concurrent multiple myeloma and HLH.

The presented case is unique in many respects. First of all, only 2% of myeloma patients are younger than 40 years of age at diagnosis. Secondly, HLH was until recently not commonly associated with multiple myeloma or its treatment, although this syndrome is becoming increasingly frequent with the administration of newer treatment schemes [37,38,39]. Despite the fact that the patient did not receive such medication, his clinical and laboratory status (Table 1), as well as his response to immunoregulatory treatment were remarkable. Moreover, necrotizing laryngitis is expected to manifest in the context of immunosuppression, although it is regarded as a rare entity currently. Its etiology may be immune-associated (e.g., systemic lupus erythematosus, granulomatosis with polyangiitis), malignancy-associated (e.g., extranodal NK/T-cell lymphoma, radiotherapy, bevacizumab), or—in the majority of the cases—infectious [40,41,42]. Moreover, although a series of bacteria, viruses, and fungi may cause necrotizing laryngitis, to our knowledge, this is the first case report that demonstrated pan-drug-resistant (PDR) *Acinetobacter baumannii* as the causative microorganism. Bacterial invasion of the larynx from this particular pathogen is a curious occurrence, yet this might be attributed to hematogenous spread, as *Acinetobacter baumannii* was repeatedly isolated from consecutive blood cultures.

In the literature, there are two further reports of patients with necrotizing laryngitis with possible involvement of *Acinetobacter* to its pathogenesis. The first case was that of “black larynx” [25]. This patient’s management required tracheostomy, surgical debridement, and a three-agent regimen against *Acinetobacter* consisting of meropenem, tigecycline, and colistimethate sodium. The other case was of a 43-year-old with poorly-managed diabetes, who had acute epiglottitis, which was ascribed to *Acinetobacter baumannii* infection. Disease course was further complicated by necrotizing fasciitis and descending mediastinal abscess due to *Peptostreptococcus anaerobius* [43].

*Acinetobacter baumannii* was introduced as a major nosocomial pathogen just 40 years ago and this phenomenon has assumed epidemic proportions since [44]. Currently, this microbe is frequently isolated from intensive care unit patients in many Eastern European and Asian countries, while in Turkey and Greece it ranks as the most common isolate [45]. Another worrisome issue is the rising antibiotic resistance. While *Acinetobacter baumannii* is inherently resistant to many antibiotic classes, its ability to quickly develop new resistant strains is a cause for consideration, as up to 95% of *Acinetobacter baumannii* isolates from Greek intensive care unit patients are resistant to meropenem [46]. Carbapenem resistance mechanisms may be summarized into two categories, namely antibiotic-inactivating enzymes (carbapenemases) and reduced antibiotic entry. A report from the Centers for Disease Control and Prevention in 2019 mentioned carbapenemase production in 83% of carbapenem-resistant *Acinetobacter baumannii*. These genes are usually chromosomally encoded (e.g., *OXA-23-like*, *OXA-24/40-like* and *OXA-58-like* oxacillinases) or rarely, plasmid-acquired (e.g., *KPC*, *IMP*, *NDM*, *VIM*, *OXA-48-like*) [47]. The latter are more frequently associated with other major Gram-negative pathogens and further exacerbate the problem of antibiotic resistance. Carbapenem resistance is further supplemented by the fact that, *Acinetobacter baumannii* have fewer and narrower porin channels than other bacteria. Point mutations of genes encoding porins may lead to a decrease in their number, while other mutations lead to efflux pump upregulation [48]. In addition, prior meropenem exposure, as occurred in our patient, is a probable risk factor for multi-drug resistant *Acinetobacter baumannii* [49]. With respect to this, the Infectious Diseases Society of America has included carbapenem-resistant *Acinetobacter baumannii* in its list of the seven most impactful antimicrobial-resistant hospital-associated pathogens and has published related treatment guidelines. Furthermore, most *Acinetobacter* isolates in Greece display at least intermediate susceptibility to tigecycline and almost half of them are resistant to colistin [46]. Colistin is a cationic polypeptide that interacts with the negatively charged lipopolysaccharide (LPS) molecules of the cell membrane. It displaces positively charged ions (Mg^2+^, Ca^2+^), which disrupts cell membrane integrity. Resistance to colistin is primarily mediated by LPS modification (PEtN moiety addition to lipid A, mutations or overexpression of *pmrCAB*, *eptA*, presence of plasmid-mediated *mcr* genes, the overall charge of the outer membrane may be altered and thus colistin may not interact with lipid A of LPS, inactivation of the LPS biosynthetic pathway) and the overexpression of efflux pumps [50]. Colistin resistance is a major consideration, therefore, and agent choice should be guided by susceptibility tests; however, for pan-drug-resistant strains, options are limited to a combination of high-dose ampicillin/sulbactam, meropenem, tigecycline, and colistin [46,51,52,53]. Recently two novel agents, namely the siderophore cephalosporin Cefiderocol and Sulbactam-Durlobactam, a combination of a beta-lactamase inhibitor with beta-lactam activity plus a non-beta-lactam/beta-lactamase inhibitor, have shown promise as a potentially life-saving alternative for patients unresponsive to previous antibiotic regimen [54,55]. The significance of these agents is further highlighted by the fact that this group of patients are co-colonized by other major nosocomial pathogens. This feature has been associated with a number of adverse outcomes. Therefore, the incorporation of antibiotics with efficacy against *Pseudomonas*, *Klebsiella*, and *Acineobacter* is advisable. Nonetheless, this situation once more underscores the all-important subjects of prevention and antibiotic stewardship.

In conclusion, this is the first case of necrotizing laryngitis due to PDR *Acinetobacter baumannii* in a patient with concomitant first diagnosis of multiple myeloma and HLH. Patients with hematologic diseases, who receive immunosuppressive or immunomodulatory treatment may develop uncommon infections with atypical manifestations, such as severe necrotizing laryngitis. Tissue culture is important to establish the infectious cause and determine the optimal antimicrobial drug therapy. In the appropriate context, multi-drug-resistant bacteria should be suspected and opportunistic infections should be ruled out. Such patients require multi-disciplinary management in specialized centers.

## Figures and Tables

**Figure 1 microorganisms-12-01382-f001:**
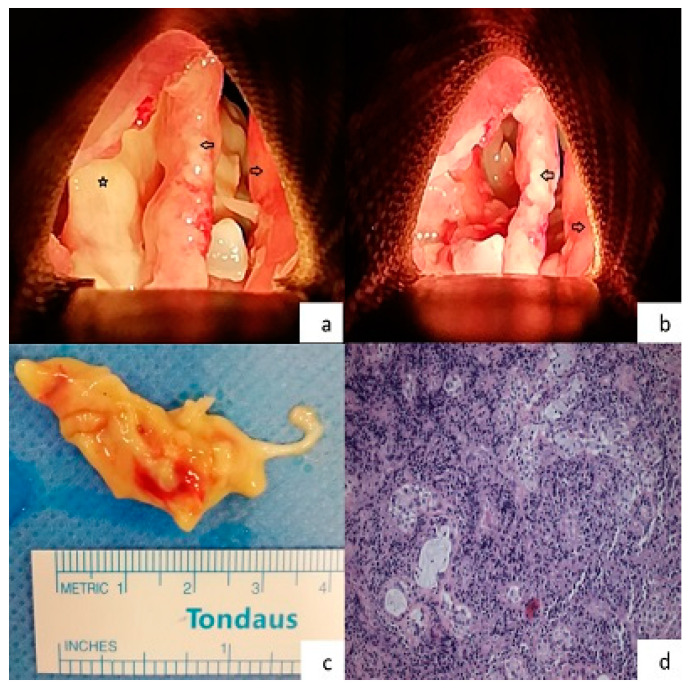
(**a**) Intraoperative image of the patient’s larynx during the panendoscopy procedure. Both of the true vocal cords are intact. The yellowish area beneath the left true vocal cord demonstrates the necrotic area (Right arrow = right true vocal cord, Left arrow = left true vocal cord, Star = necrotic area, Orientation: upwards is the anterior commissure), (**b**) intraoperative image of the patient’s larynx during the panendoscopy procedure. The necrotic area that extended from the vallecula to the subglottic area was removed (Right arrow = right true vocal cord, Left arrow = left true vocal cord, Orientation: upwards is the anterior commissure), (**c**) the specimen of the tissue that was removed from the necrotized area, (**d**) histologic specimen from supraglottic area (hematoxylin and eosin stain—magnification ×40). Necrotized tissue with extensive neutrophilic infiltrate. Some seromucinous glands are preserved in the stroma.

**Table 1 microorganisms-12-01382-t001:** Laboratory values upon admission to the HDU and on the 16th day of treatment in the same department (immediately prior to tracheostomy).

	Admission	Day 16
Complete blood count		
Hemoglobin (g/dL)	5.9	6.9
Hematocrit (%)	17	20
Red blood cells (×10^6^/μL)	1.9	2.3
White blood cells (10^3^/μL)	2.5 (75% PMNs)	0.2
Platelets (10^3^/μL)	60	17
Biochemical panel		
Mg^++^ (mg/dL)	1.8	2.3
K^+^ (mmol/L)	4.3	3.2
Na^+^ (mmol/L)	126	132
Aspartate aminotransferase (U/L)	286	22
Alanine aminotransferase (U/L)	778	30
Creatine phosphokinase (U/L)	189	68
Lactate dehydrogenase (U/L)	1833	164
Gamma-glutamyl transferase (U/L)	70	63
Serum creatinine (mg/dL)	7.82	1.93
Urea (mg/dL)	234	73
Total serum protein (g/dL)	7.8	6.6
Serum albumin (g/dL)	2.0	2.5
Glucose (mg/dL)	111	105
Total bilirubin (mg/dL)	2.58	1.05
Direct bilirubin (mg/dL)	2.48	-
Ca^++^ (mg/dL)	5.1	7.4
Urate (mg/dL)	7.6	6.2
PO_4_^+++^ (mg/dL)	4.4	2.8
Alkaline phosphatase (U/L)	59	117
Inflammation markers		
High-sensitivity C-reactive protein (mg/dL)	10.84	2.71
Procalcitonin (μg/L)	31	0.22
Erythrocyte sedimentation rate (mm/h)	>140	-
Ferritin (ng/mL)	>10,500	-
Coagulation panel		
Prothrombin time (s)	23	16.6
Activated partial thromboplastin time (s)	55.7	44.7
Fibrinogen (g/L)	3.7	4.9
D-dimers (g/L)	35.5	0.91

**Table 2 microorganisms-12-01382-t002:** Demographic characteristics, causative microorganisms, and outcome of cases with non-necrotizing and necrotizing laryngitis among patients with hematologic disease.

Demographic Characteristics		Non-Necrotizing Laryngitis	Necrotizing Laryngitis
Number of cases		38	14
Median age (range, years)		41	21
		(1.5–74)	(2–66)
Gender	Female	17/38 (45%)	9/14 (64%)
	Male	21/38 (55%)	5/14 (36%)
Hematologic Disease	AML	13/38 (35%)	3/14 (21%)
	ALL	5/38 (13%)	4/14 (30%)
	Lymphoma	10/38 (26%)	2/14 (14%)
	CLL	3/38 (8%)	0/14 (0%)
	MM	2/38 (5%)	0/14 (0%)
	HLH	0/38 (0%)	2/14 (14%)
	Aplastic anemia	3/38 (8%)	1/14 (7%)
	Other	2/38 (5%)	2/14 (14%)
Epiglottitis		21/38	3/14
Pathogens	Total cultured pathogens	44	23
	Gram-positive cocci *	7/44 (16%)	3/23 (13%)
	Other Gram-positive bacteria **	4/44 (9%)	2/23 (9%)
	Gram-negative bacteria ^#^	4/44 (9%)	6/23 (26%)
	Viruses (*VZV*)	1/44 (2%)	0/23 (0%)
	Fungi	26/44 (59%)	12/23 (52%)
	None identified	2/44 (5%)	0/23 (0%)
Fungal pathogens	Patients with fungal infections	22/38 (58%)	10/14 (71%)
	*Candida* spp.	16/22 (73%)	1/10 (10%)
	*Aspergillus* spp.	4/22 (18%)	8/10 (80%)
	Zygomyces (e.g., mucor)	2/22 (9%)	2/10 (20%)
	*Histoplasma capsulatum*	1/22 (5%)	0/10 (0%)
Outcome	Airway securement	16/32 (50%)	11/13 (85%)
	Clinical Improvement	21/37 (57%)	9/14 (64%)
	Death	16/37 (43%)	5/14 (36%)

Abbreviations: AML = acute myeloid leukemia, ALL = acute lymphocytic leukemia, CLL = chronic lymphocytic leukemia, MM = multiple myeloma, HLH = hemophagocytic lymphohistiocytosis; * *Streptococcus pyogenes*, *Staphylococcus aureus*, *Streptococcus pneumoniae*, *Enterococcus* spp.; ** *Nocardia* spp., *Actinomyces*, *Corynebacterium* spp.; ^#^
*Moraxella catarrhalis*, *Enterobacter clocae*, *Stenotrophomonas maltophilia*, *Serratia marcescens*, *Pseudomonas aeruginosa*, *Klebsiella pneumoniae*, *Eikenella corrodens*, *Prevotella melaninogenica*.

**Table 3 microorganisms-12-01382-t003:** Characteristics of 14 patients with hematologic disease and necrotizing laryngitis.

Patient (Ref)	Age (Years)/Sex	Hematologic Disease	Laryngoscopic Findings	Histologic Findings	Pathogen(s)	Airway Management	Outcome of Laryngeal Disease (LD)
1 [6]	21/F	Acute lymphocytic leukemia	Necrotizing epiglottitis	Necrotic tissue with fungal organisms and gram-negative bacilli, angioinvasion and vascular thrombosis, destruction of epiglottis	*Aspergillus flavus* *Klebsiella pneumoniae*	Trach	Death related to LD
2 [7]	16/F	Hemophagocytic lymphohistiocytosis due to *EBV*	Hemorrhagic, necrotic mass surrounding the base of the epiglottis	Fibrin clot containing gram-negative bacilli and gram- positive cocci	*Pseudomonas aeruginosa* (R to piperacillin)	Trach	Death related to LD
3 [8]	66/F	Procainamide-induced neutropenia	Necrotizing epiglottitis	Diffuse necrosis and superficial candidiasis	*Staphylococcus aureus Escherichia coli*	Int	Improvement
4 [9]	6/F	Acute lymphoblastic leukemia, cord blood transplantation	Edema of subglottic mucosa	Necrotic cells and fungal hyphae	*Aspergillus fumigatus*	Int	Improvement
5 [10]	46/F	Acute monoblastic leukemia, hematopoietic cell transplantation	nad	Mucosal necrosis of glottis and sublottis, numerous fungal hyphae	*Aspergillus fumigatus*	Nad	Death related to LD
6 [11]	19/F	Aplastic anemia	White to yellow necrotic tissue surrounding the subglottic area	Fungal hyphae	*Aspergillus fumigatus*	Trach	Death related to LD
7 [12]	2/M	Acute lymphocytic leukemia	Pale plaque on the right vocal cord	Inflammatory and necrotic cells, fungal hyphae	*Aspergillus fumigatus*	Trach	Improvement
8 [13]	5/M	Hemophagocytic lymphohistiocytosis due to infection	Necrotizing supraglottitis	Necrotic tissue with numerous bacterial colonies and lymphoid tissue undergoing degeneration	*Enterococcus faecalis*, *Eikenella corrodens*, *Prevotella melaninogenica*, *Neisseria* spp.	Int	Improvement
9 [14]	3/F	Acute lymphocytic leukemia	White membranes on epiglottis	Necrosis	*Corynebacterium diptheriae* non-toxinogenic	Int	Improvement
10 [15]	46/M	Lymphoma Burkitt (*HIV*)	Necrotizing laryngotracheitis	Fungal hyphae	*Mucor*	Trach	Death not related to LD
11 [16]	58/M	CNS DLBCL	White pseudomembranes on vocal folds, supraglottis and cricoid cartilage	Necrotic tissue and fungal hyphae	*Aspergillus fumigatus*	No	Improvement
12 [17]	46/F	Acute Myeloid Leukemia	Necrotizing laryngitis, complete layngotracheal separation	nad	*Aspergillus flavus* *Candida albicans* *Candida dubliniensis Saccharomyces cerevisiae Corynebacterium* spp. *Enterococcus faecium*	Trach	Improvement
13 [18]	59/M	Acute Myeloid Leukemia	Laryngeal edema, left vocal cord lesion ball	Necrotic tissue partly covered by metaplastic squamous epithelium, numerous septate fungal hyphae	*Aspergillus fumigatus*	No	Improvement
14 [19]	52/F	Allogeneic hematopoietic cell transplantation	Necrotic laryngopharyngitis	nonseptate hyphae, invasive mucormycosis	*Mucor*	Trach	Improvement

Abbreviations: LD = laryngeal disease, F = female, M = male, Trach = Tracheostomy, Int = Intubation, *EBV = Ebstein–Barr virus*, R = resistant, nad = not available data, *HIV = human immunodeficiency virus*, CNS = central nervous system, DLBCL = diffuse large B-cell lymphoma.

## Data Availability

No new data were created or analyzed in this study. Data sharing is not applicable to this article.

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
