# Peer review of "Necrotizing Laryngitis in Patients with Hematologic Disease: The First Case-Report Due to PDR Acinetobacter baumannii and Literature Review"

_microorganisms, 2024, doi:10.3390/microorganisms12071382_

Round 1
Reviewer 1 Report
Comments and Suggestions for Authors
Dear colleagues, thank you for the interesting paper submitted to Microorganisms journal. It is an excellent paper.
There are some aspects that need to be clarified/corrected
in Abstract section you are stating that your patient is 37 years old [row 19] and in the body of the article it is stated 38 years [row 54] - please clarify age
In rows 76-77 there is a potentially confusing statement regarding chronology ["chemotherapy consisting of etoposide and dexamethasone was started on the 75 second day of his admission"]. It needs clarification because in rows 60-62 you are stating "A few days thereafter, erythroderma appeared and the possibly culpable agent lenalidomide was discontinued. However, the patient soon developed fever up to 40oC without rigors, dyspnea, abdominal pain and dysuria." It is second day in HDU (High Dependency Unit)? Please correct/nuance statement. Same question for row 78 regarding 5th day of admission... in HDU?!
It is mandatory to have a single time-line in your entire paper because chronology could be confusing... please address values in rows 79,80, 82 and 83
In table 1 there are also issues
page 2 in CBC section you are stating "Hemoglobin (gr/dL)" - unit is "g/dL"
page 3 in Coagulation panel] you have inserted "Activated partial thromboplatin time" - please correct to thromboplastin...
Table 2 in page 5 in Pathogens section is needed a correct alignment because it is not evident what number is corresponding to each pathogen.
Same malalignment in Fungal pathogen section
table 3 - in patient 10 reference 15 there is a typo error : " 10 [15[[ "
In patient 11 another typo error : "bra-nes on vocal"
In patient 12 another : "yngotra-cheal"
In row 217 there are some symbols with no understandable meaning. Please clarify : "in a retropharyngeal abscess και descending necrotizing mediastinitis"
In discussion section you can add a comment on potential role of initial meropenem treatment in selecting such a pan-drug resistant Acinetobacter infection as stated by Sophonsri and co [Sophonsri A, Kelsom C, Lou M, Nieberg P, Wong-Beringer A. Risk factors and outcome associated with coinfection with carbapenem-resistant Klebsiella pneumoniae and carbapenem-resistant Pseudomonas aeruginosa or Acinetobacter baumanii: a descriptive analysis. Front Cell Infect Microbiol. 2023 Oct 16;13:1231740. doi: 10.3389/fcimb.2023.1231740. PMID: 37908764; PMCID: PMC10613969.] that are stating : "More coinfected patients were severely debilitated, had prior carbapenem exposure (37% vs 13%, p<0.001)"
You can also add some considerations in discussion about treatment section regarding cefiderocol that seems to be the only rescue therapy for complete non-responders :
1. Karruli A, Migliaccio A, Pournaras S, Durante-Mangoni E, Zarrilli R. Cefiderocol and Sulbactam-Durlobactam against Carbapenem-Resistant Acinetobacter baumannii. Antibiotics (Basel). 2023 Dec 14;12(12):1729. doi: 10.3390/antibiotics12121729. PMID: 38136764; PMCID: PMC10740486.
2. Dan MO, TÇŽlÇŽpan D. Friends or foes? Novel antimicrobials tackling MDR/XDR Gram-negative bacteria: a systematic review. Front Microbiol. 2024 May 10;15:1385475. doi: 10.3389/fmicb.2024.1385475. PMID: 38800756; PMCID: PMC11116650.
Please revise References section entirely because citation style is not appropriate.
Let's have an example. Reference 3 in your paper is : Siegel, R.L., et al., Cancer statistics, 2023. CA Cancer J Clin, 2023. 73(1): p. 17-48.
And PubMed style is : Siegel RL, Miller KD, Wagle NS, Jemal A. Cancer statistics, 2023. CA Cancer J Clin. 2023 Jan;73(1):17-48. doi: 10.3322/caac.21763. PMID: 36633525.
Please add ethics committee approval number for your study, if existing.
Comments on the Quality of English LanguageMultiple typo errors and malalignment in tables should be addressed.
Reviewer 2 Report
Comments and Suggestions for Authors
Dear Authors,
The article, with a study focused on a secondary pathology in immunocompromised pathologies such as multiple myeloma, brings to the fore the detailed presentation of a particular case through the evolution of laryngeal necrosis.
As the authors declare, only two cases of necrotizing laryngitis in patients with multiple myeloma are highlighted in the literature, which makes this study special.
The detailed presentation of this case is remarkable, the authors present the evolution of the young patient studied, with hemophagocytic lymphohistiocytosis in which the presence of Acinetobacter baumannii was identified.
As the authors present, detailing this case is useful and necessary for understanding and broadening the horizon of knowledge in the pathology associated with the present bacterium Acinetobacter baumannii, considered to be the most pathogenic worldwide and which often requires an association of several antibiotics with mechanisms different action to oppose the resistance of the pathogen to different classes of antibiotics.
From this perspective, the article is extremely interesting by bringing to the fore the various pathologies associated with the presence of the superbacterium Acinetobacter baumannii.
Reviewer 3 Report
Comments and Suggestions for Authors
1. Words of Table2 are asymmetric, please modify.
2. In the case description section, the authors didn’t mention sputum culture. Please add the relevant data.
3. Please add the relevant chest evaluation.
4. After A. baumanii was cultured, why the authors kept the antibiotics for 2 months instead of using more effective regiment.
5. The authors didn’t mention the laryngeal reconstruction. Please add the relevant information.
Comments on the Quality of English LanguageMinor editing of English language required.
